# Atherosclerosis as Pathogenetic Substrate for Sars-Cov2 Cytokine Storm

**DOI:** 10.3390/jcm9072095

**Published:** 2020-07-03

**Authors:** Mattia Vinciguerra, Silvia Romiti, Khalil Fattouch, Antonio De Bellis, Ernesto Greco

**Affiliations:** 1Department of Clinical, Internal Medicine, Anesthesiology and Cardiovascular Sciences, Sapienza University of Rome, 00161 Rome, Italy; mattia_vinciguerra@libero.it (M.V.); romiti.1520774@studenti.uniroma1.it (S.R.); ernesto.greco@uniroma.it (E.G.); 2Department of Cardiovascular Surgery, GVM Care and Research, Maria Eleonora Hospital, 90135 Palermo, Italy; 3Department of Cardiology and Cardiac Surgery, Casa di Cura “S. Michele”, Maddaloni, 81024 Caserta, Italy; antoniodebellis@alice.it

**Keywords:** atherosclerosis, Sars-CoV-2, COVID-19, pathogenesis of Sars-CoV-2, cytokine

## Abstract

The severe acute respiratory syndrome coronavirus 2 (Sars-CoV-2) outbreak is a public health emergency affecting different regions around the world. The lungs are often damaged due to the presence of Sars-CoV-2 binding receptor ACE2 on epithelial alveolar cells. Severity of infection varies from complete absence of symptomatology to more aggressive symptoms, characterized by sudden acute respiratory distress syndrome (ARDS), multiorgan failure, and sepsis, requiring treatment in intensive care unit (ICU). It is not still clear why the immune system is not able to efficiently suppress viral replication in a small percentage of patients. It has been documented as pathological conditions affecting the cardiovascular system, strongly associated to atherosclerotic progression, such as heart failure (HF), coronary heart disease (CHD), hypertension (HTN) and diabetes mellitus (DM), could serve as predictive factors for severity and susceptibility during Sars-CoV-2 infection. Atherosclerotic progression, as a chronic inflammation process, is characterized by immune system dysregulation leading to pro-inflammatory patterns, including interleukin 6 (IL-6), tumor necrosis factor α (TNF-α), and IL-1β. Reviewing immune system and inflammation profiles in atherosclerosis and laboratory results reported in severe COVID-19 infections, we hypothesized a pathogenetic correlation. Atherosclerosis may be an ideal pathogenetic substrate for high viral replication ability, leading to adverse outcomes, as reported in patients with cardiovascular factors. The level of atherosclerotic progression may affect a different degree of severe infection; in a vicious circle, feeding itself, Sars-CoV-2 may exacerbate atherosclerotic evolution due to excessive and aberrant plasmatic concentration of cytokines.

## 1. Introduction

The severe acute respiratory syndrome coronavirus 2 (Sars-CoV-2) outbreak has been affecting different regions of the world since the end of 2019, representing a public health emergency. More than 7,960,856 cumulative cases and 434,888 deaths have been confirmed by the Center for System Science and Engineering (CSSE) at Johns Hopkins University (JHU) as of 15 June, 2020 (https://coronavirus.jhu.edu/map.html). The incubation period for Sars-CoV-2 to develop symptomatology is normally 4–5 but can extend up to 14 days [1]. Severity of infection varies from the absence of symptomatology, to fever, cough, shortness of breath, anorexia, fatigue, headache, myalgia, anosmia, and ageusia, identified as clinical criteria for case ascertainment by the Center for Disease Control and Prevention (CDC) (https://cdn.ymaws.com/www.cste.org/resource/resmgr/2020ps/interim-20-id-01_covid-19.pdf), up to the most severe cases characterized by severe pneumonia, acute respiratory distress syndrome (ARDS), and sepsis [2,3,4]. Respiratory failure, shock, and multiorgan system dysfunction describe critical illness, representing approximately 5% of cases and requiring mechanical ventilation in an intensive care unit (ICU) [5]. Pathogenesis of COVID-19 severity is not well known [6,7].

Angiotensin Converting Enzyme 2 (ACE2) is the functional receptor of Sars-CoV-2, representing the entry site to the human cells, which may be ubiquitous, although it is more often expressed by the epithelial cells of the lungs and vascular system, and myocytes [8]. Referring to Sars-CoV similarities with the ACE2 structure, it may be present on macrophages, monocytes, and lymphocytes, triggering the immunological activation [9].

The immunological response to the viral infection results in the main cause of acute organ injury: excessive activation. In patients who need ICU treatment and those with severe/critical manifestation of Sars-CoV-2, the immunological pattern is more dysregulated, being characterized by pro-inflammatory figures, leading to an abnormal and disproportionate activation of cytokine hosts in a cascade labeled a “cytokine storm” [7,9].

The causes of immune response exacerbation are largely unknown and it is not yet clear why only this small percentage of patients develops a persistent and dangerous infection, potentially leading to deterioration of their condition and eventually to death [9]. For instance, children appear to acquire severe COVID-19 infections with lower incidence, more often developing a milder clinical course and having a better prognosis [10].

A meta-analysis conducted on 53,000 infected patients in Wuhan established a few risk factors for severity of COVID-19: older age, being male, smoking, and any comorbidity. Hypertension (HTN), diabetes mellitus (DM), cardiovascular disease (CVD), cerebrovascular diseases, chronic obstructive pulmonary disease (COPD), and chronic kidney disease (CKD) show a higher significative incidence in severe cases (54.9%) than in less aggressive ones (27.6%) [11]. Except for CKD and COPD, the other comorbidity risk factors documented for severity are involved with or are a direct consequence of the atherosclerotic process, which is characterized by a well-known progression strongly related with immune system dysregulation.

Our aim was to review the pathogenetic mechanism of severe COVID-19, focusing on atherosclerotic process and describing it for profiles related to the immune system and inflammation.

## 2. Pathogenesis of COVID-19

Sars-CoV-2, as a causative agent of the novel COVID-19 respiratory infectious disease, is a single-stranded ribonucleic acid (RNA) virus characterized by high transmission human to human, mainly through respiratory droplets [12,13]. The virus primarily colonizes the lungs after an invasion of the mucous membranes and causes the viremia phase as defined by Lin et al., entering into the blood system. Typically, fever and cough are the most common symptoms detectable in this phase [6].

The high documented transmissibility of COVID-19 may be explained by the high viral load during the viremia phase and by the efficient molecular mechanism that recognizes the binding protein ACE2, allowing invasion of alveolar epithelial human cells [8,13].

The immune function of affected patients is involved in controlling the virus during the acute phase; it seems that the acute organ damage that follows the first viremia phase occurs approximately seven days after the onset of symptoms, when Sars-CoV-2 is not efficiently suppressed by immune systems. In the acute phase, exploiting ACE2 binding receptors, COVID-19 leads to worsening respiratory functionality, causing pneumonia [6]. The severity of respiratory disease and impairment varies from mild symptoms, such as cough and shortness of breath, up to acute respiratory distress syndrome (ARDS), an expression of critical illness [14]. The ubiquitous presence of ACE2 and a patient’s susceptibility may be associated with multiorgan failure, including acute myocardial infarction, causing myocarditis and kidney and liver injury leading to systemic impairment [4]. The immune system, triggered by viral replication, plays a crucial role in damaging organs during the acute phase due to the excessive activation [9].

An abnormal inflammatory response leads to exuberant amounts of cytokines and chemokines, among others. The postulated pathogenetic mechanisms involved, due to the high affinity for ACE2, are associated with the massive viral replication, targeting cells such as alveolar epithelial, endothelial, macrophagic, and lymphocytic cells, causing apoptosis and pyroptosis in the cells of the immune system.

Based on the pathogenesis of COVID-19, the role of viral induction on ACE2 may be involved in downregulation and shedding of the receptor, leading to renin–angiotensin system dysfunction and increasing vascular permeability. Deficiency of ACE2, being the enzyme involved in the modulation of the renin angiotensin aldosterone system (RAAS), is linked with a rise in angiotensin II (Ang II), leading to widely known deleterious effects such as hypertension, vascular leakage, hypertrophy, fibrosis, and, consequently, increased severity of infection [9,15]. These findings may be supported by the documented deficiency of ACE2 in African-descendent populations such as African-Americans, explaining the high incidence of severe manifestations and the rate of mortality in this community. The biological variability in the different expression of ACE2, associated in black populations with environmental selection, overexposes them to early HTN, atherosclerotic progression, and CVD, and may play in two different ways towards the Sars-CoV-2 infection.

Although deficiency of ACE2 may limit the adhesion to human cells and act potentially as a form of immunity against infection, once they acquire COVID-19, they are more susceptible to developing ARDS, sepsis, and multiorgan failure [16]. ACE2 is implicated in the inactivation of des-Arg9 bradykinin, as a component of the kallikrein–kinin molecular system, causing angioedema in the lungs due to the local vascular leakage effect [17]. Instead, the role of anti-S-protein-neutralizing antibodies in facilitating acute lung injury is controversial [9].

Laboratory results have shown a dysregulation in the immune system, showing a pro-inflammatory pattern affecting mainly the lung tissue, for large amounts of cells infiltration. COVID-19 increases the plasmatic secretion of interleukin 1β (IL-1β), interferon-γ (IFN-γ), interferon-γ-induced protein 10 kDa (IP-10), monocyte chemoattractant protein-1 (MCP-1), IL-4, and IL-10. The activation of complement system is found in the pathophysiology of ARDS, with increased plasmatic levels of C5a, and directly in the autoptic evaluation, with the high presence of C3a and C3-fragments playing a primary role and being potentially useful for an effective therapy [18].

After severe infection, the T Helper (Th) 1 lymphocytes pathway is hyperactivated, causing an excessive production of CD14++ and CD16+ inflammatory monocytes, responsible for inflammatory exacerbation and increased plasmatic concentration of IL-6 [19].

IL-6 is the key cytokine triggering increased liver production of acute-phase proteins (APPs) such as C-reactive protein (CRP) and fibrinogen, causing a hypercoagulable disease that is characterized by thrombotic and embolic events, and may predict severity of infection [20,21]. Clinical pictures of deep acute venous thrombosis and pulmonary embolism are probably related to such disorders. IL-6 is the target for therapy with tocilizumab, a monoclonal antibody that has demonstrated efficacy in severe infections and is in trial to be approved against Sars-CoV-2.

In addition to IL-6, high plasma levels of IL-2, IL-7, IL-10, granulocyte colony stimulating factor (GCSF), IP-10, MCP-1, macrophage inflammatory protein 1-alpha (MIP-1α), and TNF-α have been documented in the host, representing the well-described cytokine storm. In ICU patients, this decreases the plasmatic concentration of lymphocytes due to pulmonary aggregation and sequestration [7,9].

The pathogenetic mechanism to explain the different degrees of immune system hyperactivation able to create severe symptoms is not still clear [11]. However, analyzing statistical data from Italy and China of severe infections or related deaths is uncovering some predisposing patterns able to provoke higher susceptibility to COVID-19. In the meta-analysis conducted in Wuhan on 53,000 infected patients, severe illness and death occurred in 2% and 3.2% of cases, respectively. Being over 60 years old, being male, and the association of any comorbidity such as hypertension, diabetes, and CVD, among others, [11] represented risk factors for severity.

Data about COVID-19-related deaths in Italy (https://www.epicentro.iss.it/coronavirus/sars-cov-2-decessi-italia), as of 15 June 2020 report that 32,448 patients died, having an average age of 80 years, 58.7% were men, and only 4.1% of the cases had no comorbidities. Focusing on the prevalence of pre-existing comorbidities, the most common is hypertension, in approximately 70% of cases; diabetes, more than 30%; and coronary heart disease (CHD), 28%; followed by CKD, atrial fibrillation (AF), heart failure (HF), and COPD.

## 3. The Role of Atherosclerosis

Vascular endothelial cells provide the proper hemostatic balance, regulating inflammation, vasomotor tone, and clotting factors [22]. The ability of Sars-CoV-2 to infect endothelium was first supposed given the presence of the functional receptor ACE2 on endothelial cells surface, and then documented in engineered human blood vessel organoids in vitro [23]. While the pandemic spreads rapidly, there is mounting evidence of systemic impairment of endothelial functioning and the involvement of the cardiovascular system. Endothelial injury is caused directly by viral replication into the cells and by ACE2 downregulation, exposing them to Ang II in absence of the modulator effects of angiotensin 1–7 (Ang 1–7) [8,15]. Varga et al. [24] reported the histology exams of different organs of a small series of COVID-19 patients, two dead and one with mesenteric ischemia requiring intestine resection, revealing viral inclusion in endothelial cells and accumulation of inflammatory cells.

Endothelitis, triggering the release of cytokines and chemokines, may be determinant in the pathogenesis of severe COVID-19 infection. The aberrant production may potentially cause, due to the persistent pro-inflammatory state, coagulopathy with elevation of markers on laboratory tests, such as D-dimer, fibrinogen, antithrombin, activated partial thromboplastin time (aPTT), elevated prothrombin (PT), and, in severe manifestations, disseminated intravascular coagulopathy (DIC) [25]. Fibrin thrombus formation in post-mortem examination of lung tissue has been documented, highlighting the relevant pathogenetic role of endothelium [26]. Magro et al. [27] reported, after examining skin and lung tissues in five patients affected by respiratory and cutaneous manifestations caused by COVID-19, systemic activation of the complement pathways; they recognized deposits of terminal complement components as responsible for inflammatory thrombogenic vasculopathy leading to a clinical picture of respiratory failure different from classic ARDS.

What emerges from worldwide reports regarding clinical characteristics of patients affected by COVID-19 is an increased vulnerability due to pre-existing cardiometabolic factors. The common linkage between these different predisposing cardiovascular or metabolic conditions is represented by endothelial dysfunction; therefore, a pre-existing dysregulation may represent an independent risk factor for infection severity. In this context, atherosclerosis, which shares risk factors such as HTN and DM, among others, with severe COVID-19 infections, is a chronic inflammatory disease of the endothelium characterized by infiltration, deposit, and lipid oxidation, which activates and promotes a self-maintaining inflammatory state [28,29,30,31]. In the early stages, mechanical stress and endothelium damage enable the accumulation of several plasma lipoproteins, in particular LDL, in the sub-endothelial space where they are modified into oxidized-LDL and trigger inflammation of arterial wall. Both innate and adaptive immune systems play a crucial role in lesion formation and plaque characterization, maintaining and promoting a pro-atherogenic state.

Oxidized-phospholipids and cholesterol crystals acquiring properties of damage-associated molecular patterns (DAMPs) are recognized by toll-like receptors (TLRs) and nod-like receptors (NLRs), and activate the NLRP3 inflammasome pathway, which results in proteolytic cleavage of pro-IL-1β and pro- IL18 to mature IL-1β and IL-18 [27]. In atherosclerosis, inflammatory signaling pathways activated are TLR4/NF-κβ and the JAK/STAT, which contribute to boosting the inflammatory state, raising cytokine expression and the consequent activation of innate and adaptive immunity cells [30,32]. Therefore, the immune system is dysregulated due to chronic inflammation related to high plasmatic concentration of cholesterol, leading to a pro-inflammatory pattern and potentially increasing the susceptibility of the host to COVID-19.

IL-1β has pro-inflammatory effects inducing the expression of cytokines such as IL-6, TNF-α, IL-8, and chemokines, improving the susceptibility of macrophages to lipid deposit and enhancing local inflammation and plaque instability. IL-1β, IL-6, and TNFα are also produced by CD14++ CD16+ non-classical monocytes activated by atherosclerosis, which are strictly correlated with disease progression; similarly, complement system is associated with atherosclerotic progression, being C3 and C4 serum levels, exacerbating inflammatory responses, linked to an increased risk for CVD [31,33,34].

The T-helper1 cells were shown to be the predominant CD4+ effectors in the context of atherosclerosis, promoting disease progression due to increased expression of pro-inflammatory cytokines [35,36]. In addition, the B2 subsets of B-cells are the main activated cells, in turn exacerbating the adaptive immune response [30]. Several studies also suggested an autoimmune response [37,38] in atherosclerosis that switches regulatory T cells from an initial protective phenotype (FoxP3+) into a pathogenic one (RORγt, T-bet, Bcl-6) [36]. Similar to atherosclerotic pathogenesis, laboratory tests performed in patients affected by severe COVID-19 infections documented an increased plasmatic concentration of these cytokines and immune cells described, assigning a key role to non-classical monocytes [19,39]. This pro-inflammatory and dysregulated state may play a crucial role in increasing host susceptibility to developing a cytokine storm and worse manifestations of COVID-19 due to excessive activation of the immunological response. The Sars-CoV-2 infection may act as a trigger in these susceptible hosts in which specific inflammation pathways are already activated (inflammasome, JAK/STAT, and NF-κβ pathways) and there is a dysregulation of the immune system. Although several studies are needed, this hypothesis may partly explain the severity of infection manifestation in this class of patients. This hypothesis may be supported, besides the described role of immune system dysregulation, by the levels of tissue cholesterol characterizing atherosclerosis.

Wang et al. [40], when analyzing infectivity of Sars-CoV-2 on cultured cells, highlighted the link between capacity of adhesion, proteases activity, or endocytosis, and the levels of tissue cholesterol content rather than plasmatic concentration. They focused on different cells belonging to three macro-groups of population: young, elderly, and elderly affected by chronic inflammation, showing growing evidence of age-related infectivity. The results of those authors explain the higher frequency of the asymptomatic form of COVID-19 in children and, conversely, the higher viral replication in elderly patients affected by chronic inflammation, leading to negative outcomes. Then, vascular chronic inflammation is associated with tissue macrophages overloaded by cholesterol, increasing the possibility of acquiring a severe COVID-19 infection.

Highlighting the importance of chronic inflammation in the atherosclerotic process, studies have increasingly demonstrated the link between high levels of pro-inflammatory cytokines, such as IL-6, TNF-α, and IL-1β, and progression and instability of atherosclerosis plaque, pointing out their potential as novel target therapy [30,41,42,43]. Clinical research is focusing on agents that inhibit IL-1, IL-6, and TNFα pathways to reduce the risk of coronary heart disease and reduce adverse outcomes after injury, which may be potentially useful in COVID-19 treatment, too [29,30,42].

In the Canakinumab Anti-Inflammatory Thrombosis Outcomes Study (CANTOS) trial Ridker et al. compared canakinumab, a monoclonal antibody targeting IL-1β, with a placebo in patients with previous myocardial infarction to reduce adverse clinical outcomes. This study showed a reduction in cardiovascular events in patients who received the drug, although it also showed an increased incidence of fatal infections [44]. Anakinra (IL-1R antagonist) showed the potential to reduce the inflammatory response in acute myocardial infarction patients [45,46]. For the causal association between IL-6R-related pathways and CHD [47], targeting treatment of IL-6 receptor is a promising therapeutic approach. In this field, the impact of tocilizumab, an IL-6R monoclonal antibody, is currently being evaluated in a randomized, placebo-controlled trial [25]. Etanercept, infliximab, and adalimumab are anti-TNF-α antibodies used in rheumatoid arthritis treatment, which demonstrated significant increase in HDL cholesterol levels and endothelial function improvement in this class of patients [48,49,50,51]. Nidorf et al. successfully performed a small, prospective clinical trial with low-dose colchicine treatment to reduce cardiovascular events in patients with stable coronary disease [52], although more research is needed. Canakinumab, tocilizumab, and colchicine are actually in trial to assess safety and efficacy to treat severe manifestations in patients affected by COVID-19 (https://clinicaltrials.gov/ct2/show/NCT04362813, https://clinicaltrials.gov/ct2/show/NCT04317092, https://www.smartpatients.com/trials/NCT04322682). Anakinra was demonstrated to be safe and to be associated with clinical improvements in patients affected by moderate to severe COVID-19 manifestations [53,54].

## 4. Effects of Sars-CoV-2 on Cardiovascular System

Bonow et al. [55] postulated a hypothesis to explain the documented increased susceptibility of patients affected by coronary artery disease (CAD) and risk factors for atherosclerotic cardiovascular disease to develop adverse outcomes and death due to COVID-19 [5,11]. As described the in literature for other acute infections, the hyperdynamic circulation caused by COVID-19 in patients with predisposing factors for CAD may exacerbate the precarious balance by increasing myocardial oxygen demand, resulting in acute coronary syndrome (ACS) [56]. They speculated that ACS in infected patients may be caused by excessive cytokines leading to atherosclerotic plaque instability and rupture [55].

Albiero et al. [57] published a case report on a 70-year-old man, treated three years before by percutaneous coronary intervention (PCI) and affected by COVID-19, who presented with ACS and non-ST segment elevation myocardial infarction (NSTEMI). The coronary angiogram showed atherosclerotic spontaneous coronary artery dissection (A-SCAD) on the proximal left anterior descending (LAD) coronary artery and in-stent restenosis of a marginal branch (OM).

The progression and instability of atherosclerotic plaque is strongly associated with a rise in plasmatic concentrations of IL-6, TNF-α, and IL-1β, and protease activation, leading to plaque rupture and luminal thrombosis due to direct negative effects on the plaque protective fibrous cap [31].

The impact of Sars-CoV-2 on lipid metabolism and progression of the atherosclerotic process is confirmed by an observational study conducted to analyze long-term effects of acute Sars-CoV infection [58]. Twenty-five patients were recruited 12 years after recovery from Sars-CoV in 2003. Metabolomic analysis was performed, showing an altered lipid metabolism with a significantly increased values of free fatty acids, lysophosphatidylcholine, lysophos-phatidylethanolamine, and phosphatidylglycerol. Moreover, 44% of the patients were affected by CVD. Sars-CoV pathogenesis shows a high similarity to COVID-19, being characterized by an abnormal hyperactivation of the immune system leading to excessive amounts of cytokines [6]. Hyperactivation of pro-inflammatory patterns, mainly characterized by cytokine storm, may increase risk of restenosis in patients who underwent PCI with stent implantation due to CHD. As recently reported by Sun et al., pre-operative increased levels of IL-6, TNF-α, and IL-23 may efficiently predict risk of restenosis after PCI associated with drug-eluting stent (DES) implantation [59]. Besides ACS, the cardiovascular system may be involved in patients affected by heart failure due to hemodynamic decompensation and, in a small percentage of cases, acute myocarditis may occur without prior evidence of CVD due to ACE2 presence on cardiac myocytes, potentially leading to chronic dilated cardiomyopathy [55,60,61].

Hospitalized patients who developed myocardial injury were described by Shi et al. [62] and Guo et al. [63], highlighting similar characteristics. Evidence of myocardial injury was characterized by increased plasmatic levels of high-sensitivity troponin I (TnI). Clinically, patients with an elevation of TnI are older and characterized by a high prevalence of HTN, DM, CAD, and HF. A more severe systemic inflammation was documented, including higher plasmatic concentration of leucocytes and CRP among others, leading to a more complicated respiratory picture with a higher incidence of ARDS requiring assisted ventilation than in patients without evidence of myocardial injury. In particular, Shi et al. [62] documented that in a total population of 416 hospitalized patients studied with confirmed diagnosis of Sars-CoV-2, 82 patients (19.7%) had evidence of myocardial injury, resulting in a mortality rate in this group of 51.2%, significantly higher than in patients without elevation of TnI (4.5%). Similarly, Guo et al. [63] studied 187 patients with confirmed laboratory diagnosis; the 27.8% with evidence of myocardial injury were characterized by a higher in-hospital mortality rate (57.6%) compared with patients not affected (8.9%). In this report, patients with pre-existing CVD as a comorbidity but without TnI elevation showed a worse outcome than patients with no comorbidity, with a mortality rate of 13.30%.

Potentially, the long-term effects of cardiovascular system involvement, mainly due to hemodynamic changes, atherosclerotic progression, and resulting increased risk of thrombosis due to COVID-19, may directly impact left ventricular systolic function and increase retrograde pressure on the right cardiac chambers leading to decompensation. Increased incidence of deep vein thrombosis (DVT) events due to abnormal blood clotting may cause more pulmonary embolism events and pulmonary hypertension.

## 5. Conclusions

The proposed pathogenetic correlation of atherosclerosis and Sars-CoV-2 infection is simplified and summarized in Figure 1. Atherosclerosis, a chronic inflammatory disease, may be an ideal substrate for the high viral replication capacity of Sars-CoV-2 in human cells, further leading to hyperactivation of pro-inflammatory patterns due to immune system dysregulation. Most likely, the level of atherosclerotic progression influences the severity of COVID-19 in susceptible patients, causing different degrees of excessive amounts of immune system cells, cytokines among others, mainly involved in the organ damage. The aberrant inflammatory response, as in a vicious cycle feeding itself, may lead to atherosclerotic progression, increasing risk of instability and rupture. However, methodological studies focused on this topic are necessary to fortify these suggestions.

## Figures and Tables

**Figure 1 jcm-09-02095-f001:**
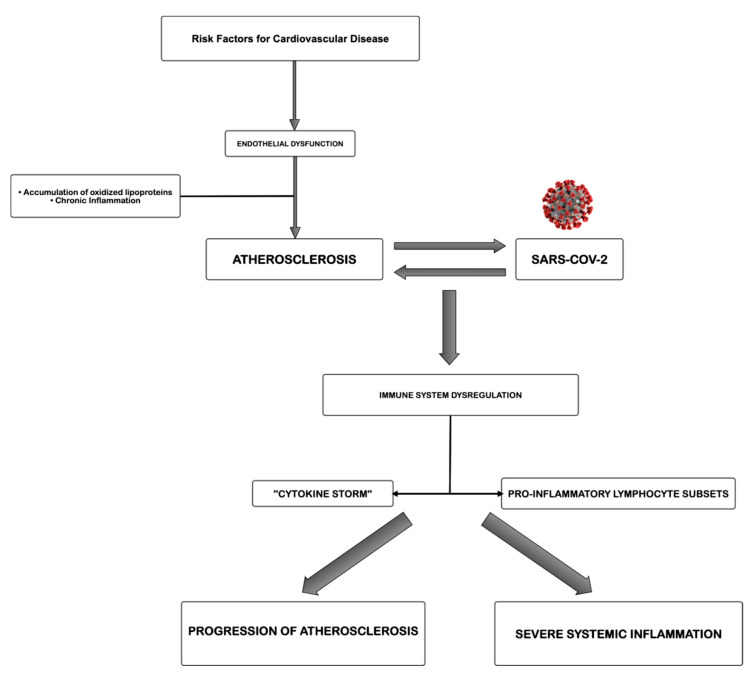
Proposed pathogenetic correlation between atherosclerosis and severe acute respiratory syndrome coronavirus 2 (Sars-CoV-2).

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
