# Peer review of "Atherosclerosis as Pathogenetic Substrate for Sars-Cov2 Cytokine Storm"

_jcm, 2020, doi:10.3390/jcm9072095_

Round 1

Reviewer 1 Report

The manuscript titled "Atherosclerosis as pathogenetic substrate for Sars-Cov2 ‘‘cytokine storm’’" is a review article. The authors have attempted to describe very well the pathogenic mechanisms induced by the cytokine storm initiated due to COVID-19 and its detrimental effects in cardiovascular well being. While the authors have explained in great deal about the signaling cascades during cytokine storm, the manuscript falls short of sufficient compelling evidences to support their hypothesis. Furthermore, more research is needed in this field and connection of atherosclerosis to COVID-19 might be accurate but it's again too early assumption. The authors have, although cited few research papers supporting the fact but in order to be considered for publication of a well written review article, more evidences and background information with well detailed diagrams are needed to support the manuscript. The figures presented by the authors needs very detailed improvement as they are of very low quality. As such I would recommend substantial corrections and rewriting of the manuscript. In the current format, the manuscript is not deemed suitable for publication.

Author Response

Dear Editor and Reviewers,

We thank you and Reviewers for revising the document and for your suggestions in order to improve the quality of our paper. We have written it more than one month ago and considering rapidly increase in evidences and knowledge regarding COVID-19 pathophysiology, we have decided to follow your recommendations. In particular:

  1. a) As suggested by Reviewer 1, we have make more clear the connection between atherosclerosis and Sars-CoV-2, reporting evidences about viral capacity to infect endothelium triggering pro-inflammatory patterns and the deducible susceptibility giving from pre-existing chronic inflammation; further in the paragraph concerning role of atherosclerosis we have discussed the article reporting the association between age-related infectability and tissue cholesterol content;
  2. b) We have removed Figure 1 which described racial and gender differences, as suggested by Reviewer 2 and simplified Figure 2 making it an intuitive and basic diagram, as suggested by Reviewer 1;
  3. c) We have removed the consideration about Tocilizumab in the conclusion and added some lines about common implication of anti-inflammatory therapy delaying atherosclerotic progression and the actual studies on efficacy and safety to treat COVID-19 manifestations, as suggested by Reviewer 2;
  4. d) We have removed smoking from description of common risk factors, as suggested by Reviewer 2;
  5. e) We have added symptoms in the description of clinical presentation of COVID-19, as suggested by Reviewer 2;
  6. f) We have added a briefly consideration on children infectability by Sars-CoV-2, as suggested by Reviewer 2;

Thank you for taking time to evaluate our paper: “Atherosclerosis as pathogenetic substrate for Sars-CoV-2 ‘‘cytokine storm’’ and for your kind consideration.

Reviewer 2 Report

This manuscript by Vinciguerra et al. presents a review of the literature on the link between CoV2-SARS infection and atherosclerosis, which the authors describe as a "cytokine storm substrate". They briefly describe the pathogenicity of the viral infection, followed by a presentation of atherosclerosis and the currently proposed (anti-inflammatory) treatments, highlighting the link between inflammation and atherosclerosis. The authors finally present their hypothesis, highlighting a probable correlation between atherosclerosis and the high viral replication of the virus thus inducing a high severity of COVID-19 disease. Anti-inflammatory treatments such as anti-IL6 are then suggested by the authors to limit the severity of the disease

This review seems to me to be rather relevant during this period, and allows a better understanding of various points currently being discussed by the scientific community. This synthesis is particularly interesting and allows us to understand from a different angle (through atherosclerosis) the risks, treatments and evolution of COVID19.

Major comments

1- I am not particularly confident in the proposed Figure 1, what do the green arrows mean? Why a black arrow? The group of people is not clear, I presume it corresponds to the general population, but is it relevant? I am not sure about the usefulness of this figure, it does not help in understanding the text. Either it should be removed or reworked with new captions and a better explanation. 

2- Concerning the conclusion of the article the proposal to carry out an anti-IL6 treatment (Tocilizumab) is consistent with the article, however, this drug is currently used to treat certain severe forms of COVID19 infection without their efficacy having yet been clearly evaluated in this indication. A few citations on the corresponding clinical trials are missing here. 

Minor comments

1-The symptoms of COVID are presented as fever and cough during the acute phase, this is indeed the case, but other important symptoms such as fatigue, aches and pains, headaches, loss of smell or taste should be mentioned. 

2- There are a lot of questions about the smoking part right now, and I'm not sure that smoking is a risk factor in the context of the SARS-CoV-2. In fact, several studies seem to point to potential protection. Although it's not clear, I think it would be preferable to remove the reference to smoking at this time (https://www.qeios.com/profile/2178). 

3- In viral symptomatology, it might be interesting to talk about the case of children, as you quote at the end of the paper the study by Wang et al. to make the link between age, chronic inflammation and asymptomatic patients. A sentence on the fact that children are not or only slightly affected and talking about asymptomatic cases at the beginning of the manuscript could add coherence to this part. This remark is just a suggestion.

Author Response

(The authors gave the same response as above.)

Round 2

Reviewer 1 Report

I would like to thank the authors for carefully addressing all the comments and concerns. I believe the manuscript is fit for acceptance.